# Knowledge, Attitude and Practices of Self-Medication Including Antibiotics among Health Care Professionals during the COVID-19 Pandemic in Pakistan: Findings and Implications

**DOI:** 10.3390/antibiotics12030481

**Published:** 2023-02-28

**Authors:** Zia Ul Mustafa, Shahid Iqbal, Hafiz Rahil Asif, Muhammad Salman, Sehar Jabbar, Tauqeer Hussain Mallhi, Yusra Habib Khan, Tiyani Milta Sono, Natalie Schellack, Johanna C. Meyer, Brian Godman

**Affiliations:** 1Discipline of Clinical Pharmacy, School of Pharmaceutical Sciences, Universiti Sains Malaysia, Gelugor 11800, Penang, Malaysia; 2Department of Pharmacy Services, District Headquarter (DHQ) Hospital, Pakpattan 57400, Pakistan; 3Department of Medicine, Tehsil Headquarter Hospital (THQ), Choa Saiden Shah, Chakwal 48800, Pakistan; 4Department of Medicine, Quaid-e-Azam Medical College, Bahawalpur 63100, Pakistan; 5Institute of Pharmacy, Faculty of Pharmaceutical and Allied Health Sciences, Lahore College for Women University, Lahore 54000, Pakistan; 6Department of Clinical Pharmacy, College of Pharmacy, Jouf University, Sakaka 72341, Saudi Arabia; 7Department of Public Health Pharmacy and Management, School of Pharmacy, Sefako Makgatho Health Sciences University, Ga-Rankuwa 0208, South Africa; 8Saselamani Pharmacy, Saselamani 0928, South Africa; 9Department of Pharmacology, Faculty of Health Sciences, University of Pretoria, Pretoria 0084, South Africa; 10South African Vaccination and Immunisation Centre, Sefako Makgatho Health Sciences University, Ga-Rankuwa 0208, South Africa; 11Department of Pharmacoepidemiology, Strathclyde Institute of Pharmacy and Biomedical Science (SIPBS), University of Strathclyde, Glasgow G4 0RE, UK; 12Centre of Medical and Bio-Allied Health Sciences Research, Ajman University, Ajman P.O. Box 346, United Arab Emirates

**Keywords:** hospitals, healthcare workers, COVID-19, self-medication, antibiotics, Pakistan, antimicrobial resistance, AWaRe classification

## Abstract

Since the emergence of COVID-19, several different medicines including antimicrobials have been administered to patients to treat COVID-19. This is despite limited evidence of the effectiveness of many of these, fueled by misinformation. These utilization patterns have resulted in concerns for patients’ safety and a rise in antimicrobial resistance (AMR). Healthcare workers (HCWs) were required to serve in high-risk areas throughout the pandemic. Consequently, they may be inclined towards self-medication. However, they have a responsibility to ensure any medicines recommended or prescribed for the management of patients with COVID-19 are evidence-based. However, this is not always the case. A descriptive cross-sectional study was conducted among HCWs in six districts of the Punjab to assess their knowledge, attitude and practices of self-medication during the ongoing pandemic. This included HCWs working a range of public sector hospitals in the Punjab Province. A total of 1173 HCWs were included in the final analysis. The majority of HCWs possessed good knowledge regarding self-medication and good attitudes. However, 60% were practicing self-medication amid the COVID-19 pandemic. The most frequent medicines consumed by the HCWs under self-medication were antipyretics (100%), antibiotics (80.4%) and vitamins (59.9%). Azithromycin was the most commonly purchase antibiotic (35.1%). In conclusion, HCWs possess good knowledge of, and attitude regarding, medicines they purchased. However, there are concerns that high rates of purchasing antibiotics, especially “Watch” antibiotics, for self-medication may enhance AMR. This needs addressing.

## 1. Introduction

Since the emergence of the coronavirus disease of 2019 (COVID-19) in China, Pakistan has been at considerable risk of the virus due to border sharing, trade and travel ties with China through land, sea and air [1,2,3]. After reporting the first positive case of COVID-19 in Pakistan on 26 February 2020, comprehensive measures were introduced by the government to reduce its spread, similar to other countries [1,4,5,6,7,8]. These measures included a country wide lockdown, which resulted in the closure of markets, educational/religious institutes and marriage/banquet halls, as well as the cancellation of religious congregations and ceremonies and public and private sport festivals [4,8,9]. However, despite these initiatives, cases of COVID-19 continued to rise in Pakistan, with a substantial number of positive cases reported in different waves of the pandemic. This rise was exacerbated by concerns with poverty and unemployment in Pakistan with continued lockdown measures as well as social distancing and other measures not being strictly followed [4,8,10].

Alongside lockdown and other measures, there were designated COVID-19 wards among secondary and tertiary care hospitals throughout Pakistan providing care to patients hospitalized with COVID-19 [4]. Up to 25 January 2023, more than 1.5 million people had the virus in Pakistan, with over 30,000 deaths out of a population of over 200 million [11].

Healthcare workers (HCWs) including physicians, nurses and pharmacists are regarded as first-line employees in the fight against COVID-19. Consequently, they are more likely to contract severe COVID-19 than the overall population [12,13,14,15,16]. HCWs typically treat COVID-19 patients admitted to hospitals in emergency units, general and isolation wards, intensive care units (ICUs) and critical care units (CCUs) [17,18,19,20]. By mid-May 2021, more than 115,000 HCWs had lost their lives globally after being infected with COVID-19 [21]. Local media reports in Pakistan revealed that 14,627 HCWs had been infected throughout Pakistan by 22 March 2021, with 143 deaths [22]. Not surprisingly, the increased chance of infection, coupled with the pressures placed on HCWs by the pandemic including working in high-risk environments and longer hours, adversely impacted on the mental and physical health of HCWs across countries [23,24,25,26]. A similar situation was seen in Pakistan [27,28].

With the progression of pandemic, many antimicrobials including hydroxychloroquine, ivermectin, remdesivir and azithromycin along with vitamins and other supplements to boost the immune system had been proposed to prevent or treat COVID-19 [29,30]. However, the vast majority of re-purposed medicines had little or no clinical benefit for patients, whilst increasing morbidity, mortality and costs [31,32,33,34,35,36,37,38,39,40]. The exception was dexamethasone among ventilated patients in hospital with COVID-19 [41]. The prescribing of antimicrobials including hydroxychloroquine as well as ivermectin was fueled by social media, often without input from healthcare professionals, negatively impacting on the health of patients [42,43,44,45]. This called for a greater cognizance of evidence-based medicine, as well as greater scrutiny regarding clinical trial design, when different approaches have been suggested for the treatment of patients with COVID-19, endorsed by the approaches taken in the WHO Solidarity Trial and the UK Recovery Trials [31,34,35,46]. The same situation was seen with respect to the appreciable prescribing and dispensing of antimicrobials, including antibiotics, across sectors to treat patients with COVID-19 despite limited evidence of bacterial or fungal secondary or co-infections [4,29,47,48,49,50,51,52,53,54]. This included self-medication with antimicrobials across countries, especially among low- and middle-income countries (LMICs) [39,40,54,55]. This includes Pakistan, with extensive self-purchasing of antimicrobials and currently limited measures in place to reduce this [56,57,58]. Such practices are a concern as this will increase antimicrobial resistance (AMR) rates unless addressed, increasing morbidity, mortality and costs [59,60,61,62,63].

The threat of COVID-19 itself may well lead to increased self-medication among HCWs in view of the pressures on them [64,65]. This is a challenge if unnecessary consumption of medicines results in increased costs and morbidities due to adverse drug reactions (ADRs) and increased drug–drug interactions alongside the masking of certain symptoms and delays in seeking professional help [29,66,67]. Systematic reviews and other studies have suggested that vitamins, including Vitamins C and D, can play a role in the prevention and management of patients with COVID-19; however, there can be concerns with the level of evidence [68,69,70,71,72,73].

Within Pakistan, rising AMR rates are a concern, including both the multi-drug resistant (MDR) and extensively drug resistant (XDR) bacteria, acknowledged in the recent National Action Plan to reduce AMR [74,75,76,77]. Self-medication with antibiotics among the general population, as well as among HCWs, is one of the contributing factors to rising AMR rates including within Pakistan [56,58,78,79,80,81,82,83,84]. Published studies have highlighted that self-medication with antibiotics is common among the general population in Pakistan for a number of reasons. These include a lack of time to visit doctors, the poor socioeconomic status of patients, the convenience of community pharmacies/drugs stores, previous successful experiences with antibiotics and antibiotics freely available at pharmacies/drugs store without a prescription [56,58,85,86,87]. A key concern is the extent of dispensing of “Watch” and “Reserve” antibiotics with their increasing potential for resistance [56,88,89]. This is in addition to the extensive prescribing of antibiotics among hospitalized patients in Pakistan before and during the pandemic enhancing AMR rates [4,50,90,91,92,93].

To date, numerous studies have been conducted among the general population [94,95,96], university students [97], and medical students [98], to determine their self-medication knowledge, behaviors and rationale. To the best of our knowledge, however, no study evaluating HCW self-medication habits during the COVID-19 pandemic has been published. Since HCWs including community pharmacists play a crucial role in the prevention and management of COVID-19, including enhancing vaccination uptake [65,99,100,101,102], it is imperative to assess their knowledge and attitudes towards self-medication. This includes the purchasing of antibiotics without a prescription. In this context, we conducted a multi-center, cross-sectional study with the primary objective of evaluating the awareness and practices of self-medication, including antibiotics, among HCWs in the Punjab Province in Pakistan amid the current COVID-19 pandemic.

## 2. Results

The investigators reached out to 1450 HCWs in an attempt to recruit them for the study. Overall, 1173 HCWs subsequently agreed to participate in the study, which gave a response rate of 80.9%. As far as the category of HCWs are concerned, the principal categories were physicians (31.9%), nurses (28.0%), health technicians (17.3%) and pharmacists (11.8%). The characteristics of the study sample are documented in Table 1. There was a preponderance of HCWs <40 years old (72.4%) and females (51.7%). Overall, 9.5% of participants had obtained a post-graduation or specialization in their field, with the highest number of the study HCWs providing services in secondary care institutions (48.6%).

Figure 1 depicts the participants’ responses to all the knowledge items regarding self-medication. Overall, 86.4% were aware of what self-medication was and 81.7% gave correct responses to the question assessing the safety of self-medication. Overall, 83.1% agreed that all medicines (herbal, over-the-counter agents, prescription medicines) have adverse effects, and all the study participants knew that a physician’s help must be sought in case of any adverse effects from self-medication rather than managing these effects on their own. Taken together, the study participants appeared to have a good understanding of self-medication and the associated risks (correct rate 78.1% to 100% for all items; Figure 1).

There was a significant difference in the knowledge score among both occupation and education categories (Table 2). The findings of a post hoc analysis of knowledge scores between the different HCW occupational and education categories are shown in Table 3. There was no significant difference in the knowledge score between medical doctors and pharmacists; however, both had better knowledge scores than nurses and physiotherapists. Participants with a bachelor’s degree or specialization had significantly better knowledge scores than HCWs with only a diploma.

Figure 2 illustrates HCW’s attitudes towards self-medication. Approximately 37% considered self-medication as a part of self-care, and many HCWs were of the opinion that they can diagnose and manage many different diseases by themselves and did not require any advice from a medical specialist. Approximately 62% reported that they do not recommend self-medication to others. Furthermore, an appreciable proportion (strongly agreed 7.6%, agreed 40.1% and neutral 17.3%) of our sample believed they can successfully self-medicate as they had access to all the healthcare information. As shown in Table 2, there was no significant difference in the attitude scores among the different demographic variables.

Overall, 774 HCWs (60%) reported they were self-medicated during the COVID-19 pandemic. The reasons for self-medication are shown in Table 4. Approximately 42% were taking medication in order to prevent them from catching the virus, whereas 20.7% self-medicated because they suspected they had COVID-19 symptoms. Nearly 28% self-medicated to treat their colds and influenza.

The details of the different types of medicines used for self-medication are shown in Figure 3. The top three most commonly purchased medicines were antipyretics (100%), antibiotics (80.4%) and vitamins (59.9%).

Further details of individual medicines used for self-medication are given in Table 5. A total of nine antibiotics were used for self-medication, with azithromycin being the most commonly purchased antibiotic followed by amoxicillin and co-amoxiclav. A concern is that an appreciable number of purchased antibiotics were “Watch” antibiotics (Figure 4).

A number (9%) of the study participants also reported consuming ivermectin. Out of the 568 HCWs who reported using vitamins, the majority were taking a Vitamin C and calcium combination to boost their immunity. Herbal supplements use was observed in 9% of the study population.

## 3. Discussion

We believe this is the first study conducted in Pakistan to ascertain the extent of self-medication among HCWs during the recent COVID-19 pandemic. Encouragingly, the response rate was high at 80.9%, and the majority of participating HCWs had good knowledge regarding self-medication. Most participants knew that herbal, over the counter (OTC) and prescription medicines have adverse effects and that consultation among physicians is necessary when adverse events occur. More than one-third of HCWs considered self-medication as a part of self-care and they felt they could diagnose different diseases by themselves. These findings are similar to a study from Saudi Arabia among medical and pharmacy students where most participants had good knowledge of self-medication [103]. However, physicians and pharmacists had better knowledge of self-medication in our study compared with nurses and other HCWs. This is similar to the findings of Akande-Sholabi et al. (2021) in Nigeria, where pharmacy students possessed good knowledge of self-medication versus nursing students [104]. This may be due to the fact that pharmacists generally have good knowledge of medicines, and they are considered to be custodians of medicines. This is reflected by the fact that community pharmacists are often the first healthcare professionals (HCPs) that patients consult with in the community before seeing a physician, further endorsed by the recent COVID-19 pandemic [105,106,107,108,109].

Pharmacists should typically have good knowledge of the adverse effects of medicines and the importance of their appropriate use. Moreover, HCWs having graduate or master degrees had better knowledge compared with diploma holders in our study.

Our study showed that 60% of the HCWs surveyed practiced self-medication during the COVID-19 pandemic, with common reasons for self-medication being the prevention of an infection followed by cold/flu symptoms and suspecting they have COVID-19 infections. These findings are in line with a previous study among medical students from Pakistan, Medical and Pharmacy students in Iran [98,110], as well as among the general population in Peru during the COVID-19 pandemic and globally in pooled studies [55,111]. In addition, this was also the case among HCWs in Nigeria (54.6–89.3%) [104,112]. However, this is in contrast to another study in Nigeria, where only one third of HCWs surveyed practiced self-medication during the COVID-19 pandemic [64], and in Ethiopia (22.7%) [113], with similar findings among pooled studies of HCWs across multiple countries (32.5%) [111].

Paracetamol was consumed by all HCW participants in our study along with variable use of Vitamin C plus calcium (40.6% of HCWs) and multivitamins (32.8%). The high use of paracetamol reflects the typical symptoms of patients, with the variable use of vitamins probably reflecting some of the controversies surrounding their use, including concerns with the evidence base [68,69,70,71,72,73]. However, this needs further investigating before any definitive conclusions can be drawn.

Of concern was the appreciable number of HCWs (80.4%) stating that they were self-medicating with antibiotics despite very limited bacterial co-infections or bacterial secondary infections seen even among patients hospitalized with COVID-19 [4,50,52,53]. In addition, there was appreciable purchasing of “Watch” versus “Access” antibiotics despite regulations limiting such activities. This behavior may relate to the overall stress of potentially contracting COVID-19 and/or concerns with its consequences [114]. These findings are similar to a study among HCWs in Nigeria (71.2%) [112], and also among the general population with COVID-19 in Iran as well as among Medical and Pharmacy students in Iran and Sudan during the pandemic [110,115,116]. However, this is different to HCWs in Togo, where there was only limited self-purchasing of azithromycin (1.4%) to manage their COVID-19 symptoms [117]. This is a concern, as their overuse, particularly of “Watch” antibiotics, will increase AMR rates, and the associated impact on morbidity, which is already a key issue in Pakistan [59,76,77,80,118,119,120]. There are also concerns with self-medication with ivermectin in view of its lack of effectiveness in these patients coupled with concerns with side-effects [42,121,122,123].

Potential ways forward among HCWs, especially in LMICs, include ensuring appropriate knowledge among healthcare students during their studies in order that they are fully conversant regarding antibiotics and antimicrobial stewardship (AMS), as well as helping to undertake antimicrobial stewardship programs (ASPs) on graduation. This includes knowledge of the AWaRe classification of antibiotics, suggested antibiotics for treating a range of infections in both ambulatory and hospital care with the emphasis on “Access” antibiotics, and the implications for excessive prescribing of “Watch” and “Reserve” antibiotics [88,89,124]. This is because there have been concerns with the level of knowledge of these key issues among students and healthcare professionals in LMICs [125,126,127,128,129,130,131,132,133].

In view of these findings, there is a need to ensure that the curricula for new HCWs, including those on diploma courses, adequately covers all key aspects of antibiotics, AMS and ASPs. In this way, helping to ensure that HCWs are fully conversant with these key issues on graduation, which can be built upon post-qualification [80,134]. This includes all the key issues surrounding self-purchasing of antibiotics as the activities of HCWs do influence patient behavior. This is because we do know that trained community pharmacists in LMICs, combined with appropriate guidelines, do recommend alternative approaches to antibiotics for the management of self-limiting viral infections, providing direction for the future [102,105,135,136,137]. This is important going forward, with HCWs necessarily playing a key role in addressing the considerable misinformation surrounding the prevention and treatment of patients with COVID-19 as well as future pandemics [42,101].

We are aware of a number of limitations with this study. Firstly, we only conducted this study in one province in Pakistan. However, Punjab is the most populated province in Pakistan. Secondly, we only conducted the research among public institutions. This was deliberate as they treat the most patients, especially those that are most likely to purchase medicines from pharmacists and drug stores in Pakistan in view of co-payment issues. Thirdly, we generated our own questionnaire. However, this was based on multiple publications together with the experience of the co-authors. In addition, the questionnaire was piloted. Fourthly, there may be over-representation of certain HCW groups as a result of our sampling approach. In addition, there may be under-representation of others if the responders felt that giving wrong answers reflects badly on their group. Overall, despite these limitations, we are confident of our findings in view of the methodology used and the number of HCWs taking part. As a result, we believe our findings will provide direction to the authorities in Pakistan as they move forward with the NAP to reduce AMR rates in Pakistan.

## 4. Materials and Methods

### 4.1. Study Design

A cross-sectional study design was employed in this study, and data was collected from HCWs over a four-month period from May–August 2022 using a convenient sampling technique. All HCWs included in the study worked in the public sector among primary, secondary and tertiary healthcare institutions in six districts (Pakpattan, Sahiwal, Faisalabad, Bahawalpur, Okara and Chakwal) of Punjab Province. We chose Punjab Province for this study in view of its population size within Pakistan, i.e., accounting for more than half of the population of Pakistan. Furthermore, similar research has been conducted in this province, with this research following on from existing research studies undertaken by the co-authors [4,50,125,133,138].

### 4.2. Data Collection Tool and Data Procedure

The data collection tool used in our survey was developed from previous studies combined with the considerable experience of the co-authors as well as observing local practices [55,103,110,113,114,139]. We have used this method before when developing context-specific data collection forms [101,131,140,141,142].

A pilot study was conducted among twenty-five HCWs across a range of different occupations in the districts Pakpattan and Sahiwal to assess the reliability and validity of the study instrument. The overall value of the study instrument fell within the acceptable range of internal consistency.

Following the suggested recommendations from the participants in the pilot study, the final version of the study instrument (Appendix A) incorporated the following five sections:Section I: Comprised nine questions relating to the demographic details of the study population. This included their age, gender, residence, marital status, designation, level of education, working department in the hospital, institution type and working experience.Section II: Consisted of seven questions relating to their knowledge of self-medication. Each questions had three options: “yes”, “no” and “don’t know”, designed to assess the extent of their knowledge about the appropriate use of antibiotics.Section III: Consisted of seven questions to extract information about the attitude of self-education among HCWs. Study participants were requested to select one option in a 5-item Likert scale, with the 5 items ranging from strongly agree to strongly disagree, which is typically seen in such scales [143,144,145].Section IV: Consisted of three questions to assess the prevalence of self-medication among the HCWs taking part, their reasons for self-medication and any subsequent improvement in any of their symptoms as a result of self-medicationSection V: Enlisted commonly used types of medicines for self-medication in viral diseases such as COVID-19. These included antibiotics, antiallergic, antipyretics and supplements potentially purchased by the HCWs. Potential choices of antibiotics were based on the experience of the co-authors supplemented with observing local practices. Hydroxychloroquine was not included with high-profile studies, showing no benefit and potential harm [31,37,146].

The data collection team comprised physicians, hospital pharmacists, and pharmacy/laboratory technicians. They subsequently visited the various public health facilities during the study period to enlist support for completing the questionnaire. The study instrument was distributed among the HCWs of these hospitals and they were requested to provide their responses. Completion was entirely voluntary, and informed consent was given before the questionnaires were distributed. The completed study questionnaires were subsequently collected by the investigators, with each questionnaire anonymous to help reduce any misinformation and potential bias in their replies.

### 4.3. Inclusion and Exclusion Criteria

Our survey included HCWs currently working in public sector primary, secondary and tertiary healthcare facilities in six districts in Punjab Province during the study period (May–August 2022).

Our survey excluded HCWs who were working in other cities in Punjab Province or in the private sector in these six districts during this period. HCWs that did not have enough time to participate in the study were also excluded from our survey.

### 4.4. Data Analysis Including Statistical Analysis

We entered and analyzed the data in the SPSS^®^ version 22 for Microsoft Windows. Continuous variables were expressed as median and interquartile range, whereas categorical variables were expressed as numbers and percentages. Non-parametric inferential statistical techniques were used to compare knowledge and attitude scores between demographic variables (Mann–Whitney U test and Kruskal–Wallis Test). Chi-Square test was used to determine the difference of self-medication practices between demographic variables. Statistical significance was taken as an alpha value of less than 0.05.

Purchased antibiotics were analyzed in accordance with the WHO AWaRe classification. The “access” group of antibiotics are typically considered as first- or second-line antibiotics for a range of infectious diseases. They generally have a narrow spectrum and low resistance potential. The “watch” group of antibiotics have a higher resistance potential as well as a greater side-effect profile, with the “reserve” group recommended only as a last resort and typically prioritized for any ASPs alongside agreed quality indicators [88,89]. As a result, the WHO is developing quality indicators to reduce the utilization of “Watch” and “Reserve” antibiotics in favor of greater utilization of “Access” antibiotics [147]. In addition, antimicrobials and antihistamines dispensed were classified by their ATC classification [148].

## 5. Conclusions

In conclusion, the majority of HCWs serving in public sector health facilities of different hospitals in Pakistan did possess appropriate knowledge regarding self-medication. However, there were appreciable levels of self-purchasing of antibiotics, especially those from the “Watch” category. This is a concern that needs to be urgently addressed in Pakistan to reduce rising levels of AMR, especially given the influence of HCWs in Pakistan and beyond in preventing and treating COVID-19.

Key future activities start in the universities to ensure adequate coverage of antibiotics, the AWaRe classification and book containing suggested first and second line treatments for multiple infections and their importance, as well as AMS and ASPs. Coverage of ASPs includes providing adequate input to key tasks to ensure HCWs are confident about implementing them post-qualification. Such educational activities need to be continued post-qualification to help improve the appropriateness of future antimicrobial use as well as address misinformation regarding infectious diseases and their treatment, which has been especially prevalent during the current pandemic.

Community pharmacists have a key role to play in Pakistan in the future to reduce AMR, especially given the existing high levels of self-purchasing of antibiotics among the general population. We have observed that trained pharmacists in some LMICs are directing patients with self-limiting infectious diseases towards alternative treatments that do not include antibiotics, and this should be encouraged in Pakistan. We will be exploring these activities further in future studies.

## Figures and Tables

**Figure 1 antibiotics-12-00481-f001:**
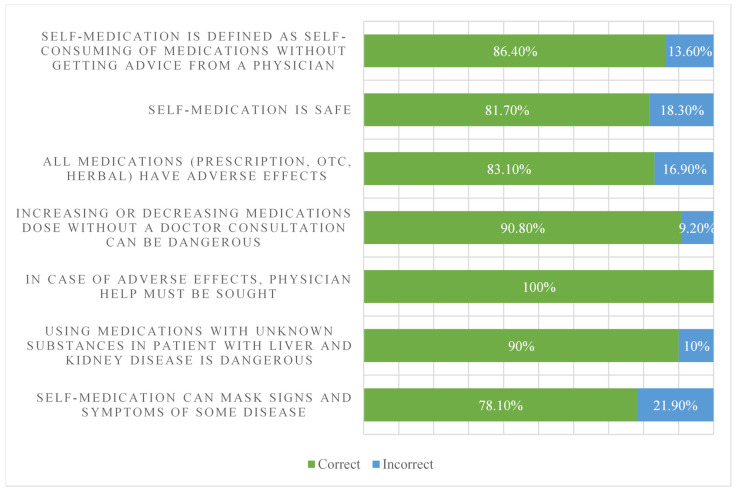
Participants’ knowledge related to self-medication. NB: N = 1173.

**Figure 2 antibiotics-12-00481-f002:**
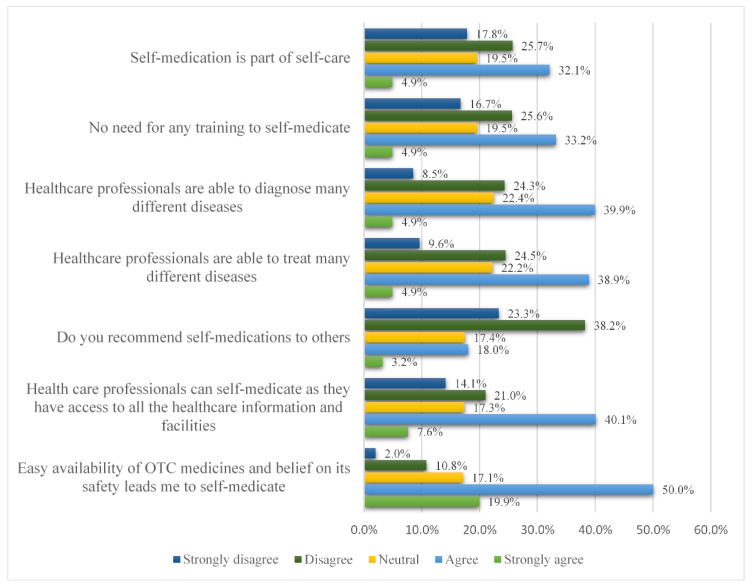
Healthcare workers’ attitudes towards self-medication. NB: N = 1173.

**Figure 3 antibiotics-12-00481-f003:**
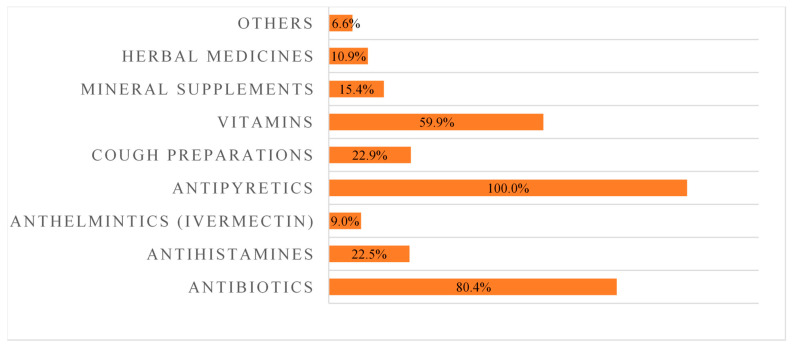
Pharmacological agents used for self-medication. NB: N = 774.

**Figure 4 antibiotics-12-00481-f004:**
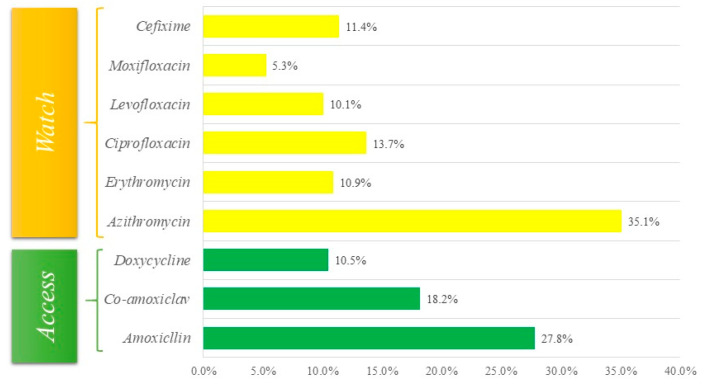
Details of antibiotics purchased by AWaRe classification. NB: “access” and “watch” antibiotics taken from the WHO AWaRe list, see Methods.

**Table 1 antibiotics-12-00481-t001:** Demographic details of the sample.

Variables	N (%)
**Age**	
<30 years	486 (41.4)
31–40 years	364 (31.0)
41–50 years	183 (15.6)
51–60 years	140 (11.9)
**Gender**	
Male	567 (48.3)
Female	606 (51.7)
**Residence**	
Urban	816 (69.6)
Rural	357 (30.4)
**Marital status**	
Single/Divorced/Widow	697 (59.4)
Married	476 (40.6)
**Occupation**	
Medical doctor	374 (31.9)
Pharmacist	139 (11.8)
Nurse	329 (28.0)
Lab technician	84 (7.2)
Physiotherapist	18 (1.5)
Health technician	203 (17.3)
Others	26 (2.2)
**Education**	
Diploma	549 (46.8)
Bachelor’s degree	512 (43.6)
Post-graduation/specialization	112 (9.5)
**Hospital**	
Tertiary	150 (12.8)
Secondary	570 (48.6)
Primary	453 (38.6)
**Experience**	
1–3 years	355 (30.3)
4–7 years	431 (36.7)
8–12 years	242 (20.6)
>12 years	145 (12.4)

NB: N = 1173.

**Table 2 antibiotics-12-00481-t002:** Comparison of self-medication-related knowledge and attitude scores among selected demographics.

Variables	Subgroups	Mean Rank		Self-Medication Practice N (%)	
		Knowledge Score	Attitude Score	Yes	No
Age	<30 years	605.69	593.70	324 (41.9%)	162 (40.6)
	31–40 years	583.30	579.92	241 (31.1)	123 (30.8)
	41–50 years	547.69	592.00	115 (14.9)	68 (17.0)
	51–60 years	583.13	575.63	94 (12.1)	46 (11.5)
	*p*-value	0.199	0.908	0.802	
Gender	Male	591.37	592.43	372 (48.1)	195 (48.9)
	Female	582.92	581.92	402 (51.9)	204 (51.1)
	*p*-value	0.645	0.594	0.805 *	
Residence	Urban	582.10	586.29	552 (71.3)	264 (66.2)
	Rural	598.20	588.62	222 (28.7)	135 (33.8)
	*p*-value	0.419	0.914	0.071 *	
Occupation	Medical doctor	776.84	570.93	239 (30.9)	135 (33.8)
	Pharmacist	731.27	601.06	80 (10.3)	59 (14.8)
	Nurse	572.62	599.87	220 (28.4)	109 (27.3)
	Lab technician	461.83	534.34	65 (8.4)	19 (4.8)
	Physiotherapist	508.11	622.50	15 (1.9)	3 (0.8)
	Health technician	264.81	618.18	137 (17.7)	66 (16.5)
	Others	241.42	482.27	18 (2.3)	8 (2.0)
	*p*-value	**<0.001**	0.236	**0.046**	
Education	Diploma	456.53	594.79	376 (48.6)	173 (43.4)
	Bachelor’s degree	694.19	583.51	327 (42.2)	185 (46.4)
	Post-graduation/specialization	736.53	564.75	71 (9.2)	41 (10.3)
	*p*-value	<0.001	0.660	0.236	
Hospital	Tertiary	575.62	609.43	98 (12.7)	52 (13.0)
	Secondary	606.91	588.08	377 (48.7)	193 (48.4)
	Primary	565.71	578.21	299 (38.6)	4154 (38.6)
	*p*-value	0.102	0.615	0.983	
Experience	1–3 years	601.99	582.00	241 (31.1)	114 (28.6)
	4–7 years	600.69	596.65	279 (36.0)	152 (38.1)
	8–12 years	545.58	585.99	152 (19.6)	90 (22.6)
	>12 years	578.72	572.23	102 (13.2)	43 (10.8)
	*p*-value	0.115	0.871	0.364	

NB * Fischer’s Exact test; Bold = statistically significant.

**Table 3 antibiotics-12-00481-t003:** Pairwise comparisons of knowledge scores in occupation and education variables.

Sample 1–Sample 2	Test Statistic	Std. Error	Std. Test Statistic	Sig.
**Pairwise comparisons of occupation**				
Others—Health technicians	23.387	65.434	0.357	0.721
Others—Laboratory personnel	220.410	70.500	3.126	0.002
Others—Physiotherapist	266.688	96.322	2.769	0.006
Others—Nurses	331.197	63.996	5.175	0.000
Others—Pharmacist	489.843	67.123	7.298	0.000
Others—Doctors	535.421	63.713	8.404	0.000
Health technicians—Laboratory personnel	197.023	40.754	4.834	0.000
Health technicians—Physiotherapist	243.301	77.256	3.149	0.002
Health technicians—Nurses	307.810	28.037	10.979	0.000
Health technicians—Pharmacist	466.456	34.584	13.488	0.000
Health technicians—Doctors	512.033	27.386	18.697	0.000
Laboratory personnel—Physiotherapist	−46.278	81.591	−0.567	0.571
Laboratory personnel—Nurses	110.787	38.402	2.885	0.004
Laboratory personnel—Pharmacist	269.433	43.414	6.206	0.000
Laboratory personnel—Doctors	315.010	37.930	8.305	0.000
Physiotherapist—Nurses	64.509	76.042	0.848	0.396
Physiotherapist—Pharmacist	223.155	78.691	2.836	0.005
Physiotherapist—Doctors	268.732	75.804	3.545	0.000
Nurses—Pharmacist	158.646	31.779	4.992	0.000
Nurses—Doctors	204.224	23.745	8.601	0.000
Pharmacist—Doctors	45.577	31.206	1.461	0.144
**Pairwise comparisons of education categories**				
Diploma—Bachelor	−237.665	19.300	−12.314	0.000
Diploma—Post-grad/Specialization	−279.999	32.571	−8.597	0.000
Bachelor—Post-grad/Specialization	−42.334	32.769	−1.292	0.196

NB: Each row tests the null hypothesis that the Sample 1 and Sample 2 distributions are the same. Asymptotic significances (2-sided tests) are displayed. The significance level is 0.05.

**Table 4 antibiotics-12-00481-t004:** Respondents’ practices related to self-medication amid the COVID pandemic.

	N (%)
Self-medication during COVID-19 *	
Yes	774 (66.0)
No	399 (34.0)
Reason for self-medication? **	
Cold or flu	216 (27.9)
COVID-19 prevention	324 (41.9)
Suspected COVID symptoms	160 (20.7)
COVID-19 positive	36 (4.7)
Consuming regularly	19 (2.5)
Miscellaneous	19 (2.5)
Symptom improvement **	
All symptoms improved	40 (5.1)
Many symptoms improved	326 (42.0)
Some symptoms improved	295 (38.0)
One symptom improved	89 (11.4)
No improvement	28 (3.6)

NB: * N = 1173; ** N = 774.

**Table 5 antibiotics-12-00481-t005:** Medications self-administered to treat suspected COVID-19.

Pharmacological Class	Individual Medicine	Overall N (%)
Antibiotics	Amoxicillin (J01CA04)	215 (27.8)
	Co-amoxiclav (J01CR02)	141 (18.2)
	Azithromycin (J01FA10)	272 (35.1)
	Ciprofloxacin (J01MA02)	106 (13.7)
	Levofloxacin (J01MA12)	78 (10.1)
	Moxifloxacin (J01MA14)	41 (5.3)
	Cefixime (J01DD08)	88 (11.4)
	Erythromycin (J01FA01)	84 (10.9)
	Doxycycline (J01AA02)	81 (10.5)
Antihistamines	Ebastine (R06AX22)	49 (6.3)
	Fexofenadine (R06AX26)	56 (7.2)
	Cetirizine (R06AE07)	85 (11.0)
Anthelmintics	Ivermectin (P02CF01)	70 (9.0)
Vitamins	Multivitamins	254 (32.8)
	Vitamin C + Calcium	314 (40.6)
Mineral supplements	Zinc supplement	118 (15.2)
Antipyretics	Paracetamol	774 (100.0)
Cough preparations	Cough preparations	177 (22.8)
Herbal medicines	--	84 (10.9)
Others	--	51 (6.6)

NB: N = 774.

## Data Availability

Further data are available upon reasonable request from the co-authors.

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
