# Peer review of "Knowledge, Attitude and Practices of Self-Medication Including Antibiotics among Health Care Professionals during the COVID-19 Pandemic in Pakistan: Findings and Implications"

_antibiotics, 2023, doi:10.3390/antibiotics12030481_

Round 1

Reviewer 1 Report

I found this article interesting for the readers and followed the journal Antibiotics’ scope. I don’t have any major comments as this article has enough data and is well written with proper discussion.

I would recommend the article be published in Antibiotics after minor corrections. 

The author needs to address the following comments/corrections.

 1.     Recheck the summation of percentage s in few cases it is not 100% (e.g Table 1, entries 1,4 and 5).

2.     Table 1: Needs foot notes (for all tables and figures).

3.     All numerical data should be in the same format (e.g instead of 94 write 94.0 etc.)

4.     Table 2: Entry 1 (age) check alignment.

5.     Section 1 (page 15)  already in the supplementary.

6.  The author should correct the format of references wherever needed (e.g Year Bold, Volume Italic etc).

Author Response

Comments and Suggestions for Authors

I found this article interesting for the readers and followed the journal Antibiotics’ scope. I don’t have any major comments as this article has enough data and is well written with proper discussion.

Author comments: Thank you for these kind words – very much appreciated

I would recommend the article be published in Antibiotics after minor corrections. The author needs to address the following comments/corrections.

Author comments: Thank you – we hope we have adequately addressed these.

1. Recheck the summation of percentage in few cases it is not 100% (e.g Table 1, entries 1,4 and 5).

Author comments: Thank you for this. In line with multiple other publications, we have chosen to document the findings to one decimal point in the Tables – in line with comments made in the actual paper. We realise this means on occasions that the total doses not add up to 100%. However – we feel it will be inappropriate inflating (or deflating) one of the figures to achieve 100%. We hope you agree.

2. Table 1: Needs foot notes (for all tables and figures).

Author comments: Now added.

3. All numerical data should be in the same format (e.g instead of 94 write 94.0 etc.)

Author comments. Thank you for this. As you are no doubt aware – that when e.g. in Tables documenting the actual number of patients taking part (rather than e.g. to one decimal point when documenting %s) – it is customary not to use a decimal point. Consequently, we hope you agree with this.

4. Table 2: Entry 1 (age) check alignment.

Author comment: Thank you – amended.

5. Section 1 (page 15) already in the supplementary.

Author comment: Thank you – now done (we included this in the submitted manuscript as there was no opportunity to show this to you in the initial submission. Now rectified).

6. The author should correct the format of references wherever needed (e.g Year Bold, Volume Italicetc).

Author comment: Thank you – we will work with the Journal if and when our paper is accepted to ensure the current format for the references.

Reviewer 2 Report

The article entitled as Knowledge, attitude and practices of self-medication including antibiotics among health care professional during the COVID- 19 pandemic in Pakistan; findings and implications by Zia Ul Mustafa et al conducted an important research work to  investigate the Knowledge, attitude and practices of self-medication including antibiotics among health care professional during the COVID- 19.

However I have few minor comments:

1.      Add a few lines on the data collection in the abstract section.

2.      Attach the Proforma in the supplementary section as annexure and remove it from main manuscript.

3.      Concise the conclusion.

4.      Add future perspective and limitation of the study.

5.       Cite and update the introduction with updated citations.

6.       Recheck the grammar and remove the typo mistakes.

Author Response

The article entitled as Knowledge, attitude and practices of self-medication including antibiotics among health care professional during the COVID- 19 pandemic in Pakistan; findings and implications by Zia Ul Mustafa et al conducted an important research work to investigate the Knowledge, attitude and practices of self-medication including antibiotics among health care professional during the COVID- 19.

Author comments: Thank you for this summary.

However I have few minor comments:

1. Add a few lines on the data collection in the abstract section.

Author comments: Thank you - now added. 

2. Attach the Proforma in the supplementary section as annexure and remove it from main manuscript.

Author comments: Now enacted. This was included for your information as you reviewed the manuscript – now transferred to Supplementary Material. We hope this is now OK.

3. Concise the conclusion.

Author comments: Thank you – we have amended this in view of your comments regarding a future perspective. We hope this is acceptable.

4. Add future perspective and limitation of the study.

Author comments: Thank you – these were included in the submitted paper. However – we have improved the signposting and hope this is now acceptable.

5. Cite and update the introduction with updated citations.

Author comments. Thank you – we have changed citations where we can without compromising the paper. In addition, added some more recent papers to the Introduction as requested. Some citations are necessary though when discussing Pakistan – especially issues around prescribing in Pakistan, high use of antibiotics in patients with COVID-19, link between high use of antibiotics and AMR as well as endorsing our methodology. We hope this is acceptable.

6. Recheck the grammar and remove the typo mistakes.

Author comments: Thank you. The spell check changed some words from the English to the American spelling. We have also modified some of the phraseology. We hope this is now acceptable.
